The classification of movement intention through machine learning models: the identification of significant time-domain EMG features

http://orcid.org/0000-0002-6969-9062 Mohd Khairuddin Ismail 1 2
Sidek Shahrul Naim 2 snaim@iium.edu.my
http://orcid.org/0000-0002-3094-5596 P.P. Abdul Majeed Anwar 1
Mohd Razman Mohd Azraai 1
Ahmad Puzi Asmarani 2
http://orcid.org/0000-0002-6349-2028 Md Yusof Hazlina 2
1 Faculty of Manufacturing & Mechatronics Engineering Technology, Innovative Manufacturing, Mechatronics and Sports Laboratory, Universiti Malaysia Pahang , Pekan, Pahang , Malaysia
2 Department of Mechatronics, Kulliyyah of Engineering, Biomechatronics Research Laboratory, International Islamic University , Gombak, Selangor , Malaysia
Liu Pengcheng
Electronic publication date: 2021 Feb 25
Publication date: 2021
Volume: 7
Electronic Location ID: e379
Received 2020 Nov 10; Accepted 2021 Jan 7
Copyright: © 2021 Mohd Khairuddin et al.
Copyright year: 2021
Copyright holder: Mohd Khairuddin et al.
License: This is an open access article distributed under the terms of the Creative Commons Attribution License, which permits unrestricted use, distribution, reproduction and adaptation in any medium and for any purpose provided that it is properly attributed. For attribution, the original author(s), title, publication source (PeerJ Computer Science) and either DOI or URL of the article must be cited.
License URL: https://creativecommons.org/licenses/by/4.0/

Keywords: EMG, Machine learning, Feature extraction, Movement intention, Classification

Funding: Ministry of Higher Education Malaysia FRGS/1/2017/TK04/UIAM/02/12 This work was supported by the Fundamental Research Grant Scheme (FRGS) through the Ministry of Higher Education Malaysia under Grant FRGS/1/2017/TK04/UIAM/02/12. The funders had no role in study design, data collection and analysis, decision to publish, or preparation of the manuscript.

==============================
Electromyography (EMG) signal is one of the extensively utilised biological signals for predicting human motor intention, which is an essential element in human-robot collaboration platforms. Studies on motion intention prediction from EMG signals have often been concentrated on either classification and regression models of muscle activity. In this study, we leverage the information from the EMG signals, to detect the subject’s intentions in generating motion commands for a robot-assisted upper limb rehabilitation platform. The EMG signals are recorded from ten healthy subjects’ biceps muscle, and the movements of the upper limb evaluated are voluntary elbow flexion and extension along the sagittal plane. The signals are filtered through a fifth-order Butterworth filter. A number of features were extracted from the filtered signals namely waveform length (WL), mean absolute value (MAV), root mean square (RMS), standard deviation (SD), minimum (MIN) and maximum (MAX). Several different classifiers viz. Linear Discriminant Analysis (LDA), Logistic Regression (LR), Decision Tree (DT), Support Vector Machine (SVM) and k-Nearest Neighbour (k-NN) were investigated on its efficacy to accurately classify the pre-intention and intention classes based on the significant features identified (MIN and MAX) via Extremely Randomised Tree feature selection technique. It was observed from the present investigation that the DT classifier yielded an excellent classification with a classification accuracy of 100%, 99% and 99% on training, testing and validation dataset, respectively based on the identified features. The findings of the present investigation are non-trivial towards facilitating the rehabilitation phase of patients based on their actual capability and hence, would eventually yield a more active participation from them.

Introduction

Robot-assisted platforms have been introduced over the last two decades in order to complement the work of physiotherapists and have been shown to provide positive results in assisting subjects to regain their activities of daily living (ADL) (Brackenridge et al., 2016; Chen et al., 2016; Fasoli, 2016). Amongst the advantages of the utilisation of assist-based robotics therapy, it in its ability to provide autonomous training where subjects are involved in the repeated exercise of goal-directed tasks that in turn, leads to improvements in their motor function (Peternel et al., 2016; Taha et al., 2019). The importance of implementing such robot-assist platform could reduce the burden of the physiotherapist as well as facilitating active rehabilitation process of the subjects (Rahman, Ochoa-Luna & Saad, 2015; Huang, Tu & He, 2016).

Even though the use of such robot-assist platforms could significantly facilitate the rehabilitation process, nonetheless, the key to rehabilitation recovery is often dependent on the subjects’ natural capability (Rahman et al., 2015; Zimmermann et al., 2015). In order to provide assistance and keep motivating the subjects in performing the prescribed tasks programmed on the robotic platform, the selection of a suitable controller is essential. Several controllers have thus far been applied to the robot-platform to aid the patient in the movement recovery process (Rahman, Ochoa-Luna & Saad, 2015; Zimmermann et al., 2015; Proietti et al., 2016). It should be noted that in order to improve the interaction between the subject and the robotic-assisted platform, the capturing or detection of physiological motion intention data is non-trivial, as a substantial effort is required by the patient to complete the prescribed exercise. Through the identification of the motion intention, a suitable amount of assistive force could be regulated by the controller of the robotic platform to facilitate the completion of the prescribed task.

To date, different methods have been developed to detect or capture the intent of subjects. For instance, capturing intention-based triggering is one of the simplest and most widely used (Marchal-Crespo & Reinkensmeyer, 2009; Proietti et al., 2016; Bi, Feleke & Guan, 2019). This triggering signal can be either force, velocity, time thresholds, or even motion intent based on electromyography (EMG) or electroencephalography (EEG) (Krebs et al., 2003; Colombo et al., 2005; Loureiro et al., 2009; Rechy-Ramirez & Hu, 2015; Bi, Feleke & Guan, 2019).

Nevertheless, it is worth noting that triggering signals are rather artificial and does not reflect the actual intention of the subject. Conversely, the processing of EEG signals is rather complex. An effective and attractive strategy in acquiring the movement intention is surface electromyogram (EMG) signals (Bi, Feleke & Guan, 2019; Guidali et al., 2013). This method could allow the robotic system to be activated prior to the actual motion of the subject. It has been reported that EMG can provide the information of muscle activation between 40 ms and 100 ms (Winter, 1990; Lenzi et al., 2012; Kosaki et al., 2017).

In order to use the EMG signals as an input for detecting the motion intention, these signals need to be extracted first. For instance, AlOmari & Liu (2014) applied several methods to extract the features such as sample entropy, root mean square (RMS), myopulse percentage rate (MYOP), and difference absolute standard deviation value in order to classify the forearm muscle activity. Other researchers have exploited different time-domain features, namely waveform length (WL), mean absolute value (MAV), zero crossing (ZC), and slope-sign change (SSC) (Kamavuako, Scheme & Englehart, 2014) to distinguish the movement of the forearm.

In terms of the classification of the aforesaid signals, several methods have been employed. Bhattacharya, Sarkar & Basak (2017) investigated the efficacy of Subspace Discriminant (SDE) Ensemble apart from k-Nearest Neighbour (k-NN), Linear Discriminant Analysis (LDA) and Quadratic Discriminant Analysis (QDA) in classifying different set of EMG features for hand movement recognition. It was shown from the study that the SDE classifier could yield the highest classification accuracy of 83.33% for training dataset. Oskoei & Hu (2008) evaluated the performance of different SVM models (kernels) in classifying upper limb motions using EMG signals against LDA and artificial neural networks (ANN). The kernels evaluated in the study were radial-basis, linear, polynomial and sigmoid. It was shown from the study that the average accuracy for all kernels was approximately 95.57% followed by LDA and ANN.

To the best of the authors’ knowledge, although there are studies that have employed different statistical EMG time-domain features, nonetheless limited studies have highlighted the significance of the features extracted. Therefore, this article aims at identifying significant EMG time-domain features that could facilitate in the detection of motion intention from the subject that in turn will provide a seamless assistance for robotics-based rehabilitation. The remainder of this article is organised as follows: “Materials and Methods” describes the approach taken for detecting intention via the EMG signals. “Results” describes the classification methods employed on the EMG signals. “Discussion” presents the result, and “Conclusions” draws the conclusion of the present work as well as provide the future direction of the present investigation.

Materials and Methods

Participants

Ten healthy subjects (four females and six males) with no neurological or orthopaedic impairment from the Department of Mechatronics Engineering of International Islamic University Malaysia (IIUM) participated in the present investigation. The participants were between the age range of 22 and 26, with a mean and standard deviation of 24.4 years, and 3.27 years, respectively. In addition, the mean weight and height of the subjects are 68 kg and 167 cm, respectively. The experiment was conducted at a dedicated room at Biomechatronic Research Laboratory of IIUM. The experiments were conducted with the approval of the International Islamic University Malaysia Research Ethics Community (IREC) 659, and all participants gave verbal and written informed consent prior to the experiment.

Intention recognition system

Figure 1 illustrates the proposed framework for detecting the motion intention of the subject. The intention recognition system utilised in this present study consists of three main components, the PC workstation (blue-dashed box), where the main programme of the overall system is linked with the second component, that is the DAQ system (red-dashed box). The last component is the collection of sensors namely torque sensor, potentiometer as well as the EMG electrode (purple-dashed box). The complete robotic platform which consists of the aforementioned components that are used in the present study is depicted in Fig. 2.

Figure 1 Block diagram for data acquisition.

Figure 2 Robotic platform.

Data acquisition and processing

The electromyography signal is captured via the g.USBamp signal amplifier (g.tec Medical Engineering GmbH). This signal is then sent to the PC workstation for processing purpose. In order to record the EMG signal, each subject was instructed to perform three sets of flexion movements (from 0° to 45° with an interval of 2 s before further moving from 45° to 90°). These movements transpired in the sagittal plane with the upper arm being held horizontally (0°) and the subject was asked to maintain as still in the upright sitting position. During the experimental testing, each subject needs to intentionally move the robotic arm platform of the robot rehabilitation system. The electromyographic signals were sampled at 1.2 kHz. The biosignals were recorded using two disposable surface electrodes placed in a configuration suggested by the manufacturer of the signal amplifier (one on the biceps, and one on the bone which is acting as a ground) as depicted in Fig. 3. A bandpass filter was applied to filter the EMG signals between the range of 5–500 Hz. The angular motion (angle) of the prescribed movements is captured via the potentiometer that is processed via Humsoftt MF624 at a sampling rate of 0.001 Hz.

Figure 3 Electrode placement.

Feature extraction

The EMG signals were segmented into two distinct sections, namely pre-intention, and intention as depicted in Fig. 4. The intention signal is recorded based on the definition of muscle burst which transpires between 40 ms and 100 ms prior to any muscle activities (Winter, 1990; Kosaki et al., 2017). The segmentation groups are based on the aforesaid information. The features of each signal of this segmentation are extracted by using time-domain (TD) technique as it is obtained from the signals’ amplitude (Veer & Sharma, 2016; Spiewak, 2018) In each, xi is ith sample of EMG signal amplitude and L is the length of the analysis window for computing the features. The TD techniques applied in this research is described as follows (Purushothaman & Ray, 2014; Naik, 2015; Rahman, Ochoa-Luna & Saad, 2015)

Figure 4 A sample of the EMG and angular position for a single movement.

Waveform Length: Is the cumulative length of the waveform over the segment. It specifies a measure of waveform amplitude, frequency, and duration all within a single parameter (1) WL=∑i=1L−1|xi+1−xi|

Mean Absolute Value: It signifies the area under the EMG signal once it has been rectified. The MAV is used as a measure of the amplitude of the EMG signal.

(2) MAV=1L∑i=1L|xi|

Root Mean Square: The RMS denotes the square root of the average power of the EMG signal for a given period of time.

(3) RMS=1L∑i=1Lxi2

Standard Deviation: The SD of a set of data is the square root of the variance, where μ refers to the mean of the sample.

(4) SD=1L−1∑i=1L|xi−μ|2μ=1L∑i=1Lxi

Maximum amplitude: Maximum value of the EMG signal (5) Max=max|xi|

Minimum amplitude: Minimum value of the EMG signal (6) Min=min|xi|

The rationale for utilising the time-domain features is primarily due to its swift computation as the features do not require any transformation on the raw EMG data (Spiewak, 2018). Moreover, it is worth noting that the time-domain features have been widely used in both medical and engineering types of research and practices (Rahman, Ochoa-Luna & Saad, 2015).

Feature selection

The selection of features is used to reduce the number of features that do not significantly contribute to the classification of the intention of the subject. In the present study, the best feature for the classification process is attained by means of an Extremely Randomised Tree (ERT) technique. The ERT is a tree-based ensemble learning technique that combines the results of multiple de-correlated decision trees collected (Geurts, Ernst & Wehenkel, 2006). The entropy-based information gain is essentially used as the decision criteria for the significant features. Figure 5 illustrate the bar graph for each of the extracted features. It is evident from Fig. 5; the most significant features identified are the MIN and MAX, respectively.

Figure 5 Bar graph of feature importance.

Classification

Classification is the last yet essential step employed in order to achieve the objectives of the present study. In this study, five different models of classification approaches are utilised on the time-series or time-domain (TD) features extracted, namely Linear Discriminant Analysis (LDA), Logistic Regression (LR), Decision Tree (DT), k-Nearest Neighbour (k-NN), and Support Vector Machine (SVM). A total of 45,358 (15%) of the dataset was used for testing, 211,668 (70%) for training and 45,357 (15%) was used for independent testing (validation). The five-fold cross-validation technique was used on the train the models, that is on the training dataset as this method has been reported to mitigate the issue of over-fitting (Razman et al., 2019). The development of the models was carried out via Scikit-learn Python Library on Spyder 4.1.4 IDE. It is worth noting that the models are developed based on its default hyperparameters as per the utilised library.

Performance matrix

The classifier’s performance is evaluated based on its classification accuracy (CA), precision, recall, F1-score, as well as the confusion matrix. Figure 6 below demonstrates the confusion matrix as an evaluation of the predictive models.

(7) Accuracy=(TP+TNTP+TN+FP+FN)×100

(8) Precision=(TPTP+FP)×100

(9) Recall=(TPTP+FN)×100

(10) F1=2(Precision∙RecallPrecision+Recall)×100

From the equations above, the accuracy is defined as the overall classification rate and on how well the classifier predicts the classes. The precision evaluates the prediction rate of correctly predicted classes, conversely recall is the ratio of the positive classes that are correctly categorised. The F1 is the harmonic mean that is generated between precision and recall by multiplying the scale by 2. The F1 score provides the quality of the prediction especially for uneven class distribution as exhibited in the present study.

Figure 6 Confusion matrix.

Results

Figure 7 illustrates the performance of the evaluated classifier, in terms of classification accuracy (CA) based on all features, whilst Fig. 8 illustrates the CA based on selected significant features identified via the ERT algorithm. The details on the other performance measures are tabulated in Tables 1 and 2, respectively.

Figure 7 Classification accuracy for each model for all features.

Figure 8 Classification accuracy for each model for the selected features.

Table 1 Classifier models performance with all features.

Classifier model	Training time (s)	Prediction time (s)	Train (%)	Test (%)	
Prec	Recall	F1	Acc	Prec	Recall	F1	Acc	Confusion matrix	
k-NN	0.947	9.121	88	96	92	87	85	93	89	81	34,460	2,590	
6,026	2,282	
SVM	861.168	290.256	82	100	90	82	82	100	90	82	37,049	1	
8,308	0	
LDA	0.221	0.006	82	100	90	82	82	100	90	82	37,049	1	
8,308	0	
DT	3.087	0.051	100	100	100	100	98	98	98	97	36,470	580	
569	7,739	
LR	2.115	0.007	82	100	90	82	82	100	90	82	37,050	0	
8,308	0	
Classifier model	Validation (%)	
Prec	Recall	F1	Acc	Confusion matrix	
k-NN	85	93	89	81	34,460	2,589	
6,031	2,277	
SVM	82	100	90	82	37,049	0	
8,308	0	
LDA	82	100	90	82	37,049	0	
8,308	0	
DT	98	99	99	98	36,519	530	
576	7,732	
LR	82	100	90	82	37,049	0	
8,308	0	

Table 2 Classifier models performance with selected features.

Classifier model	Training time (s)	Prediction time (s)	Train (%)	Test (%)	
Prec	Recall	F1	Acc	Prec	Recall	F1	Acc	Confusion matrix	
k-NN	0.289	5.798	99	99	99	99	99	99	99	98	36,634	416	
434	7,874	
SVM	385.365	233.06	82	100	90	82	82	100	90	82	37,050	0	
8,308	0	
LDA	0.09	0.008	82	100	90	82	82	100	90	82	37,050	0	
8,308	0	
DT	0.585	0.053	100	100	100	100	99	99	99	99	36,864	186	
199	8,109	
LR	0.686	0.004	82	100	90	82	82	100	90	82	37,050	0	
8,308	0	
Classifier model	Validation (%)	
Prec	Recall	F1	Acc	Confusion matrix	
k-NN	99	99	99	98	36,676	373	
414	7,894	
SVM	82	100	90	82	37,049	0	
8,308	0	
LDA	82	100	90	82	37,049	0	
8,308	0	
DT	99	99	99	99	36,863	186	
188	8,120	
LR	82	100	90	82	37,049	0	
8,308	0	

Discussion

Tables 1 and 2 recorded the classification performance for the different classifiers evaluated with respect to all features and selected. As tabulated in Table 1, it could be seen that the DT classifier model provides the highest accuracy during training with a CA of 100% in comparison to other classifier models evaluated. Nonetheless, in the event that a new dataset is applied to this model, that is the test dataset, the classification accuracy of the model demonstrated that the model provides desirable qualities with the classification accuracy of 97%. It worth noting that, when an independent testing data is applied, the DT classifier provides a high classification accuracy of 98%. The k-NN provide the second highest training classification accuracy with 87% and a testing accuracy of 81% recorded and 81% for the independent testing (validation). Meanwhile, for other classifier models that is SVM, LDA, and LR provide the same accuracy for both training and testing demonstrated that the classifiers have reached its saturation stage with the default hyperparameters of the models. It worth noting that, even though the training time and prediction time of LDA model is fastest among others classifier, the training accuracy for this model is low compare to the DT model. Figure 7 depicts the classification accuracies across all models developed.

A similar observation could be seen in Table 2, in which only the significant features identified by ERT, that is MIN and MAX are used. It is apparent that a comparable performance is illustrated, however, it is worth noting that the training time, as well as the prediction speed of the classifier, has been improved. It is apparent from Fig. 8 that the DT and k-NN model demonstrated desirable qualities with better classification accuracies for training, testing and validation. The reduction of the features is non-trivial especially for real-time implementation as the computational expense could be significantly reduced.

The present findings have demonstrated that through selected significant features, a comparable classification accuracy is attainable. It should be noted that a better CA is reported in the present study in comparison to a similar investigation carried out by Gandolla et al. (2017). The authors employed Artificial Neural Network in classifying different motions extracted from EMG signals. The study reported an average testing CA of 76% was obtained in correctly predicting healthy subjects’ motion intention. In addition, an EMG-based motion intention classification of different reaching movement was investigated by Cesqui et al. (2013). It was shown from the study that the SVM architecture utilised of significant features identified could only yield a classification accuracy of 97.5%. Therefore, it could be clearly shown that the proposed technique employed in the present investigation could yield a reasonably well CA of motion intention.

Conclusions

This article presented an approach to detect and capture the subject’s intention through the EMG signal. From the EMG signal processed a number of features were extracted for the classification purpose, namely WL, MAV, RMS, SD, MIN and MAX. Then a feature selection method was introduced in this study in order to get the significant features by means of Extremely Randomised Tree technique. The features extracted upon investigating the feature importance, are MAX and MIN, respectively. It was demonstrated from the present investigation that the DT classifier yielded an excellent classification with a classification accuracy of 100%, 99% and 99% on training, testing and validation dataset. Future works will focus on developing a controller for a rehabilitation robot based on the output of the classifier from the EMG signal taken from the subject.

Supplemental Information

Supplemental Information 1 Subject dataset.

Click here for additional data file.

Supplemental Information 2 Coding for classification.

Click here for additional data file.

The work presented was carried out in the Biomechatronics Research Laboratory of International Islamic University Malaysia.

Additional Information and Declarations

Competing Interests

Author Contributions

Ethics

Data Availability

The authors declare that they have no competing interests.

Ismail Mohd Khairuddin conceived and designed the experiments, performed the experiments, analysed the data, performed the computation work, prepared figures and/or tables, authored or reviewed drafts of the paper, and approved the final draft.

Shahrul Naim Sidek analysed the data, prepared figures and/or tables, authored or reviewed drafts of the paper, and approved the final draft.

Anwar P.P. Abdul Majeed analysed the data, performed the computation work, authored or reviewed drafts of the paper, and approved the final draft.

Mohd Azraai Mohd Razman analysed the data, performed the computation work, authored or reviewed drafts of the paper, and approved the final draft.

Asmarani Ahmad Puzi performed the experiments, prepared figures and/or tables, authored or reviewed drafts of the paper, and approved the final draft.

Hazlina Md Yusof conceived and designed the experiments, authored or reviewed drafts of the paper, and approved the final draft.

The following information was supplied relating to ethical approvals (i.e., approving body and any reference numbers):

International Islamic University Malaysia Research Ethics Community approved this research (IREC 659).

The following information was supplied regarding data availability:

Raw data and code are available in the Supplemental Files.

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
