# Peer review of "The classification of movement intention through machine learning models: the identification of significant time-domain EMG features"

_PeerJ Computer Science, doi:10.7717/peerj-cs.379_

## Round 0.1 · original submission · Major Revisions

This manuscript studies an issue of classification of movement intention using machine learning models. It is a topic of interest in predicting movement intentions based on sEMG signals. Overall, the authors have presented some interesting results and step forward based on previous works. However, there are some issues that need to be considered and addressed as the editor and reviewers pointed out.

The contribution/novelty should be clearly stated and justified via literature. There are some typos and grammatical errors in the manuscript. It is strongly suggested that the whole work to be carefully checked by someone who has expertise in technical English writing. More detailed information on the process of data acquisition should be provided. Some basic information about the participants such as weight and height should be provided. The evaluation criteria of the designed system should be provided.

Reviewer 1 ·

Basic reporting

Title
Suggested to be revised as “The Classification of Movement Intention through
Machine Learning Models: The Identification of Significant EMG Time-Domain features”
Abstract
Line 28-29 – “the upper limb are voluntary elbow flexion and extension along the sagittal plane” Please rephrase for clarity.
Please add a sentence at the end of the abstract to reflect the importance of the findings
Introduction
L 52-53 “There are numerous controllers have been implemented to the robot-platform for assisting the subject”
Please rephrase as “Several controllers have thus far been applied to the robot-platform to aid the patient in the movement recovery process”
L 71- “In order to used” » In order to use.
A few sentences could be added to summarize the gap that the current investigation is intended to fill before stating the objecting of the study.

Experimental design

L 93- “Fig 1. illustrates the proposed framework that been utilised in this study for detecting…” Please rephrase for clarity
The methodological section should be re-organized into “Participants, Intention recognition, Data Acquisition and Processing, Feature Extraction, Feature Selection and Classification”. Some of the information, particularly with regards to the participants, intention recognition and data processing, are somewhat mixed.
Please add more detail information on the process of data acquisition during the intended and non-intended movement phases.

Add some basic information of the participants such as weight and height under the participants ‘section.

Classification
Please state how many datasets are used for training, testing as well as independent testing

Validity of the findings

The authors investigated the efficacy of electromyogram (EMG) signals in detecting as well as predicting the pre-intention and intention motion of a human subject via the employment of several machine learning models. It was reported from the finding of the study the Decision Tree-based model was able to provide a good classification accuracy of the investigated motions through considering certain features obtained from the EMG signal. Overall, the manuscript is an interesting read, and the findings could be beneficial to the stakeholders in this domain. However, there is lack of detail information particularly in the methodology section to permit replication of the study. The authors are advised to consider the following suggestion to improve the quality as well as the wider readability of the manuscript.

Discussion
Please discuss the major findings in relation to the findings reported in the literature i.e. compare the findings of the current study with the findings from the other studies to claerly highlight the contribution of the study.

Additional comments

No comment

Reviewer 2 ·

Basic reporting

No comment.

Experimental design

No comment.

Validity of the findings

No comment.

Additional comments

Some minor revisions as follows:
(1) There are many English verb tense errors throughout the paper. For example, “swab (was) applied to the skin surface in order to remove dirt and dried skin, before placing the electrode. Meanwhile, the torque and angle from the patient (are) captured by a torque sensor and potentiometer. The signal obtained by the torque and potentiometer (were) fed via…”
(2) What's the meaning of “bandpass signal which is 5 Hz for low filter and 500 Hz for the high pass signal”? Does that mean “EMG signal was band-pass filtered at 10–400 Hz”?

---

## Round 0.2 · accepted · Accept

The manuscript has been well-revised based on the two reviewers' comments and suggestions.

Reviewer 1 ·

Basic reporting

No comment

Experimental design

No comment

Validity of the findings

No comment

Additional comments

I thank the authors for revising the manuscript as suggested

Reviewer 2 ·

Basic reporting

No comment.

Experimental design

No comment.

Validity of the findings

No comment.

Additional comments

The authors have modified their work and satisfied my previous comments. Therefore, from my point of view, this paper should be accepted to be published.